# Elevated Fasting Glucose and C-Reactive Protein Levels Predict Increased All-Cause Mortality after Elective Transcatheter Aortic Valve Implantation

**DOI:** 10.3390/life13010054

**Published:** 2022-12-24

**Authors:** Gabor Dekany, Katalin Keresztes, Vince P. Bartos, Orsolya Csenteri, Sara Gharehdaghi, Gergely Horvath, Abdelkrim Ahres, Christian M. Heesch, Tunde Pinter, Geza Fontos, Sai Satish, Peter Andreka

**Affiliations:** 1Gottsegen National Cardiovascular Center, H-1096 Budapest, Hungary; 2Károly Rácz Doctoral School of Clinical Medicine, Semmelweis University, Üllői Str. 26, 1085 Budapest, Hungary; 3Florida Heart Clinic, Hallandale Beach, FL 33009, USA; 4Appolo Hospital, Chennai 700054, India

**Keywords:** TAVI, diabetes mellitus, C-reactive protein

## Abstract

Surgical aortic valve replacement in the elderly is now being supplanted by transcatheter aortic valve implantation (TAVI). Scoring systems to predict survival after catheter-based procedures are understudied. Both diabetes (DM) and underlying inflammatory conditions are common in patients undergoing TAVI, but their impact remains understudied in this patient group. We examined 560 consecutive TAVI procedures and identified eight pre-procedural factors: age, body mass index (BMI), DM, fasting blood glucose (BG), left-ventricular ejection fraction (EF), aortic valve (AV) mean gradient, C-reactive protein levels, and serum creatinine levels and studied their impact on survival. The overall mortality rate at 30 days, 1 year and 2 years were 5.2%, 16.6%, and 34.3%, respectively. All-cause mortality was higher in patients with DM (at 30 days: 8.9% vs. 3.1%, *p* = 0.008; at 1 year: 19.7% vs. 14.9%, *p* = 0.323; at 2 years: 37.9% vs. 32.2%, *p* = 0.304). The presence of DM was independently associated with increased 30-day mortality (hazard ratio [HR] 5.38, 95% confidence interval [CI], 1.24–23.25, *p* = 0.024). BG levels within 7–11, 1 mmol/L portended an increased risk for 30-day and 2-year mortality compared to normal BG (*p* = 0.001 and *p* = 0.027). For each 1 mmol/L increase in BG 30-day mortality increased (HR 1.21, 95% CI, 1.04–1.41, *p* = 0.015). Reduced EF and elevated CRP were each associated with increased 2-year mortality (*p* = 0.042 and *p* = 0.003). DM, elevated BG, reduced EF, and elevated baseline CRP levels each are independent predictors of short- and long-term mortality following TAVI. These easily accessible screening parameters should be integrated into risk-assessment tools for catheter-based aortic valve replacement candidates.

## 1. Introduction

Transcatheter aortic valve implantation (TAVI) has become the standard of care for elderly (age >75 years), inoperable, and/or high surgical-risk patients with severe symptomatic aortic stenosis (AS) [1]. Given the procedure’s reduced short-term morbidity and mortality in comparison to surgical aortic valve replacement, TAVI indications are currently expanding towards younger age and lower-risk groups. Accurate risk stratification, however, continues to be a problem in the appropriate selection of patients with severe aortic stenosis for either surgical valve replacement or transcatheter options. There are currently no universally accepted morbidity and mortality prediction models to assist in the choice of the treatment modality best suited for an individual patient. Risk scores that have been employed range from institutional instruments in development specifically for TAVI to traditional and universally accepted cardiac surgical risk assessment tools, including but not confined to the Transcatheter Aortic Valve Replacement Risk Score [2], the Society of Thoracic Surgeon-Predicted Risk of Mortality (STS PROM) [3], the European System for Cardiac Operative Risk Evaluation II (EuroSCORE II) [4], the logistic EuroScore (log ES) [5], and the Society of Thoracic Surgeons (STS) [6]. Two factors that have recently found attention as possible indicators of increased mortality risks after TAVI are the presence of diabetes mellitus (DM) and the presence of possible underlying inflammatory processes. The presence of DM is an independent predictor of increased mortality in cardiovascular disease [7]. While DM has been recognized as a risk factor for mortality after surgical aortic valve replacement [8], its effect on TAVI outcomes is less clear; publications so far have been inconsistent and at times contradictory, and they are often based on small data sets. Some authors reported no mortality difference between patients with and without DM after TAVI [9,10]. Others concluded that DM in and by itself did not affect mortality, and only diabetes requiring insulin therapy (IDDM) was independently associated with an increased mortality rate after TAVI [11]. A meta-analysis including 22 observational cohort studies suggested higher all-cause mortality after TAVI with DM [12], but a sub-group analysis of the PARTNER (Placement of AoRtic TraNscathetER Valve) trial showed the contrary: a more favorable outcome was noted among patients with diabetes [13]. The percentage of individuals carrying a diagnosis of DM in patients considered for TAVI has been reported to be between 25 and 40% [14,15], whereby the overwhelming majority of such patients suffer from Type 2 diabetes mellitus. Previous studies proved that the finding of an elevated admission blood glucose is associated with increased mortality during and after the acute phase of many conditions, including myocardial infarction, stroke, trauma, and the need for hospital admission through the emergency department for any reason [16,17,18]. Similarly, indicators of systemic inflammatory processes may be associated with increased mortality after TAVI. A recent study [19] found that the presence of diabetes mellitus, low left ventricular ejection fraction, increased baseline high sensitivity CRP, and low baseline Th2 cell counts were significant predictors of death following TAVI. Another study [20] reported that elevated baseline serum CRP levels and abnormally low baseline albumin levels were each adversely correlated with long-term survival after TAVI. Most reports on such possible predictors of mortality, however, are based on the analysis of small cohorts of patients with sample sizes at best in the low [19] to mid [20] 100 s.

Given the need for a solid fundament of data to assist in the development of better risk assessment tools for the TAVI population, we sought to assess the impact of a series of risk-factor candidates on post-TAVI mortality by retrospectively analyzing a large data set (560 TAVI patients). We examined age, body mass index, diabetes, an existing diagnosis of diabetes, fasting blood glucose, left-ventricular ejection fraction, aortic valve mean gradient, C-reactive protein levels, and serum creatinine levels. Our analysis was stratified by orally treated, noninsulin-dependent diabetes mellitus (NIDDM) and insulin-dependent diabetes mellitus (IDDM).

## 2. Materials and Methods

Five hundred and sixty consecutive elective elderly patients (mean age 80 ± 6.2 years) with severe symptomatic AS and either prohibitive or extremely high surgical risk (mean Society of Thoracic Surgeons Mortality Score 5 ± 3.8) were included in this retrospective single-center observational study that analyzed data collected from February 2012 until December 2019. The study was accepted by the Medical Research Council Scientific and Research Ethics Committee (ETT TUKEB) (IV 1562/2022/EKU), and written consent was obtained from participants enrolled in the study. All TAVI implantations had been recommended following a review of the relevant patient data by the Institutional Multi-Disciplinary Heart Team in accordance with institutional best practice guidelines. All patients were on optimal medical treatment for any concomitant diseases present. All procedures were performed at the Gottsegen National Cardiovascular Center, Budapest, Hungary. Patients with hemodynamic instability, patients who needed urgent or semi-urgent TAVI procedures, patients on vasopressor support, and patients with intra-aortic balloon pump support were excluded. Only patients suffering from type 2 diabetes mellitus were included; the cohort did not include patients with type 1 diabetes mellitus. Only self-expandable (SE) valves (30.9% Core Valve, 48.2% Evolut R, 18.2% Evolute Pro, 1.3% Lotus, 1.4% Accurate Neo Valve) were used. Valve size selection was made based on multi-slice computed tomography angiography measurements following standard guidelines. All procedures were performed through the femoral artery, and procedural details have been described in detail elsewhere [21]. The presence of DM was defined based on the patient either carrying this diagnosis in prior records, or on the patient being on prescribed antidiabetic treatment (diet, oral medications, or insulin). Further, patients without an established diagnosis of DM, but with a fasting admission glucose value of ≥11.1 mmol/L were also classified as having DM [22]. Patients with DM were subdivided into NIDDM and IDDM, the latter group including patients on insulin at least once a day. Admission fasting serum glucose (BG) and high sensitivity C-reactive protein (CRP) were measured upon hospital admission in all patients as part of our routine laboratory panel. Patients were divided into four groups depending on admission BG levels of ≤4.0, >4.–≤7.0, >7.0–≤11.1, and >11.1 mmol/L, respectively. The chosen limit of 4.0 mmol/L was based on the universally accepted definition of hypoglycemia in clinical trials. Procedural complications and outcomes data were evaluated according to the Valve Academic Research Consortium 2 criteria [23]. The primary endpoint was all-cause mortality at 30 days and 2 years post-intervention; mortality data were obtained from the national death registry. Secondary endpoints included all other complications.

## 3. Statistical Analysis

Continuous variables were presented as mean ± standard deviation (SD) and compared with the Kruskal–Wallis test. Categorical variables were introduced as percentages (counts) and compared using the chi-square or Fisher exact test. Univariate survival curves were calculated with the non-parametric Kaplan–Meier estimator and compared with the log-rank test. Using Cox regression methods, a multivariable model was generated for predictors of the 30-day, 1-year (for comparison purposes only, not a primary endpoint), and 2-year composite outcomes. In this model, no variable selection was performed. Every variable was entered into the multivariate model without univariate pre-filtering to avoid the introduction of bias. Statistical analyses were performed using the SPSS software (IBM, Armonk, NY, USA), and *p*-values less than 0.05 were considered statistically significant.

## 4. Results

In this single-center, retrospective observational study, 560 consecutive elective patients (41% male) with symptomatic AS were treated with SE transfemoral TAVI implantation, and patients were followed for 2 years after the procedure. Overall survival rates were 94.8%, 83.4%, and 65.7% at 30 days, 1 year, and 2 years, respectively.

### 4.1. Clinical Characteristics

Of the 560 patients, slightly more than one-third (36.25%) had previously been diagnosed with DM. Fifty-three patients (9.5% of the entire population) were in the IDDM group. Four patients were diagnosed with previously unknown DM, based on an admission BG >11.1 mmol/L. Patients with DM were younger and had higher BMI values compared to patients without previously known or newly diagnosed DM. The diagnosis of IDDM was associated with a significantly (*p* = 0.009) younger patient age at the time of procedure presentation compared to the average age of NIDDM patients. Compared with non-diabetics, patients with DM were more likely to be male (*p* = 0.024) and more likely to have a higher STS score (*p* = 0.005). No significant difference in pre-TAVI left ventricular ejection fraction (EF) was noted between the groups. Of note, however, patients with DM had a significantly lower peak and mean aortic valve gradient than those without DM (75.2 mmHg vs. 81.6 mmHg and 47 mmHg vs. 51.3 mmHg, all *p* < 0.05). Among patients with DM, serum creatinine levels were higher and the estimated glomerular filtration rate (eGFR) was lower compared to non-diabetics. Incidentally, higher triglyceride levels were observed in the DM group, whereas serum cholesterol levels were significantly lower among diabetic patients. Hypertension was more common among patients with DM. The presence of coronary artery disease and rates of prior percutaneous coronary intervention procedures (PCI) were similar between patients with and without DM, whereas reported prior myocardial infarction and rates of prior CABG surgery were higher among patients with DM. Clinical characteristics stratified by DM status are presented in Table 1.

### 4.2. Outcomes

Mortality rates at 30 days, one year, and two years in patients without DM were 3.1%, 14.9%, and 32.2%, respectively; mortality rates for patients with NIDDM were 8%, 19.3%, and 39.3%; mortality rates for patients in the IDDM group were 11.3%, 20.8%, and 33.9%. Kaplan–Meier survival curves are presented in Figure 1. At 30 days, a significant difference was observed between the groups by univariate analysis (*p* = 0.008), as shown in Figure 1IA,IIA. Interestingly, this initially significant survival difference diminished over time and survival curves converged at 24 months follow-up, as shown in Figure 1IB,IIB. The need for balloon post-dilatation of the newly implanted valve was less frequent among patients with DM (18.2% vs. 25.7% *p* = 0.05). No major differences were observed in other procedural factors and complications, including post-TAVI bleeding, the need for emergency cardiac surgery, the need for permanent pacemaker implantation, and the development of a peri-procedural stroke. Clinical outcomes, procedural characteristics, and technical details stratified by patients’ DM status are summarized in Table 2.

Using multivariate COX regression models, DM status, admission fasting BG, and six additional variables (age, BMI, pre-TAVI EF, pre-TAVI aortic valve mean gradient, creatinine level, and CRP) were chosen without preselection as possible independent predictors of 30-day and 2-year survival. All DM and IDDM were tested in separate models. We included preoperative EF and aortic valve mean gradient as routinely assessed markers in the TAVI population. We selected age, BMI, and creatinine levels as possibly relevant factors, though our results ultimately did not confirm their relevance for TAVI outcomes. A diagnosis of DM, however, proved to be an independent predictor of increased short-term (30-day) mortality (HR 5.37 [95% CI, 1.24*–*23.25], *p* = 0.024). When also incorporating IDDM in our model, admission fasting BG was another significant predictor of short-term mortality (HR 1.21 [95% CI, 1.04*–*1.41] for 1 mmol/L increase in BG, *p* = 0.015). All other variables analyzed failed to reach significance as predictors of the 30-day outcome (see Table 3A). At 2 years, pre-TAVI EF and CRP were significant independent predictors of long-term survival (*p* = 0.042 and *p* = 0.003, respectively, see Table 3B).

Multivariate analyses were carried out using the numerical BG value, but for univariate survival analysis patients were categorized according to their falling into one of four arbitrarily chosen BG level groups. As shown in Figure 1IIIA,IIIB, patients with elevated BG (>7–≤11.1 mmol/L) were at increased risk for 30-day and 2-year mortality compared to patients with non-elevated BG (>4–≤7.0 mmol/L) (*p* = 0.001 and *p* = 0.027). Thus, patients with non-elevated BG had significantly more favorable short-term and long-term outcomes compared to those with elevated BG.

## 5. Discussion

Short- and long-term mortality of 357 patients without known DM and 203 with DM (53 IDDM) were compared after elective transfemoral TAVI. Patients with DM had increased short-term mortality. Although mortality rates of patients with DM were also higher at 1 and 2 years, this difference was only significant at 30 days (see Table 2). Both NIDDM and IDDM were significant univariate predictors of unfavorable short-term outcomes. Patients with elevated BG (>7–≤11.1 mmol/L) were at increased risk for 30-day (*p* = 0.001) and 2-year (*p* = 0.027) mortality compared to patients with non-elevated BG (>4–≤7.0 mmol/L) (see Figure 1). After multivariate adjustments were made, we found that both an existing diagnosis of DM and an elevated admission fasting BG value were independent predictors of increased short-term (30-day) mortality (see Table 3), whereas IDDM was not. Pre-TAVI EF and CRP values were significant predictors of long-term (2-year) mortality (*p* < 0.05 and *p* < 0.01). In this patient cohort, slightly more than one-third of all patients had DM. Our rate of IDDM (26.8%) was slightly less than previously reported [24]. We furthermore found that patients with IDDM and NIDDM were younger than non-diabetics (average ages 77.5, 79.3, and 80.6 years, respectively, all *p* < 0.05), and had a higher BMI (average BMI 29.6, 29.2, 27.2 kg/m^2^, respectively, all *p* < 0.05).

Somewhat counterintuitively, obesity may be an advantageous factor for TAVI outcome [25]. Goel et al. [26] previously reported both younger age and higher BMI in patients with DM before TAVI, thereby linking two factors that favorably affect procedural outcomes to a diagnosis of DM. However, DM is also associated with accelerated atherosclerosis [27], and both, patients with IDDM and with NIDDM have higher comorbidity rates compared to non-diabetics, including a diagnosis of hypertension, a history of previous MI, or previous CABG. We hypothesize that the opposing survival impact of factors commonly associated with DM (BMI and younger age on one side, and an increased incidence of vascular disease on the other) do impact TAVI outcomes, and in fact, may help explain the apparent inconsistency in previously reported TAVI outcomes in diabetics.

Multiple previous studies proved that elevated admission BG is associated with increased mortality, both after acute cardiovascular events and in patients admitted through the emergency department, irrespective of diagnosis [16,17,18]. The impact of admission fasting BG levels on mortality in patients after TAVI has not been previously reported. Our data showed that patients with elevated admission fasting BG (>7–≤11.1 mmol/L) are at increased risk for 30-day (*p* = 0,001) and 2-year (*p* = 0.027) mortality compared to patients with non-elevated BG (>4–≤7.0 mmol/L).

Given the expanding TAVI indication towards younger age groups and lower risk profile patients, the number of diabetic patients considered for TAVI as a treatment option is rising, and more attention needs to be given to diabetes status as part of the pre-procedural risk assessment. Based on already existing outcomes data, both DM and IDDM are included in the STS risk score [6], and IDDM is included in the EuroSCORE II [4], both commonly used tools to predict survival after surgical valve replacement. Similar to fasting admission BG levels, the prognostic value of CRP regarding mortality following acute cardiovascular events is established [28], but once again, the predictive value of this biomarker in patients admitted for TAVI has not been tested in large patient cohorts. Our data showed that admission CRP is a significant independent predictor of long-term survival after TAVI (HR 1.019 [95% CI, 1.01–1.03], *p* = 0.003).

Our data also showed that patients with DM face increased short-term mortality after TAVI, and further suggest that not only a history of DM, but also an elevated admission fasting BG unrelated to the presence or absence of a patient history of DM, and, independently, CRP levels have prognostic value.

We propose that both of these values should be added as prognostic markers for the pre-operative risk assessment and treatment modality selection process for patients with severe aortic stenosis in need of an interventional or operative corrective procedure. Lastly, the question of whether good control of diabetes and the workup and possible treatment of underlying inflammatory processes pre-operatively and following TAVI might improve the long-term outcome of this procedure should be a subject of future investigations.

## 6. Study Limitations

All data are from a single center, retrospective, non-randomized, observational study, and the potential of uncontrolled confounding factors cannot be excluded completely. The studied patient cohort included only non-diabetics and patients with type 2 DM, therefore, the results may not apply to patients with type 1 DM undergoing TAVI. While the duration of DM before the procedure and information on HbA1c levels were not available for analysis in our patient cohort, the association of elevated HbA1c and higher mortality during long-term follow-up of post-TAVI patients has previously been shown by others [9]. Further, the effect of subsequent therapy for the noted hyperglycemia on admission was not assessed. Lastly, this patient cohort was confined to SE valves, placed by transfemoral access, and in elective procedures. Therefore, the results cannot necessarily be applied to patients receiving balloon-expandable valves, to those in need of alternative access routes, and critically ill patients in need of pharmacological or mechanical hemodynamic support.

## 7. Conclusions

A history of DM, elevated fasting BG, and elevated CRP values upon admission adversely affect outcomes after TAVI. We suggest that these factors should be added as prognostic markers for pre-operative risk-assessment tools for TAVI candidates.

## Figures and Tables

**Figure 1 life-13-00054-f001:**
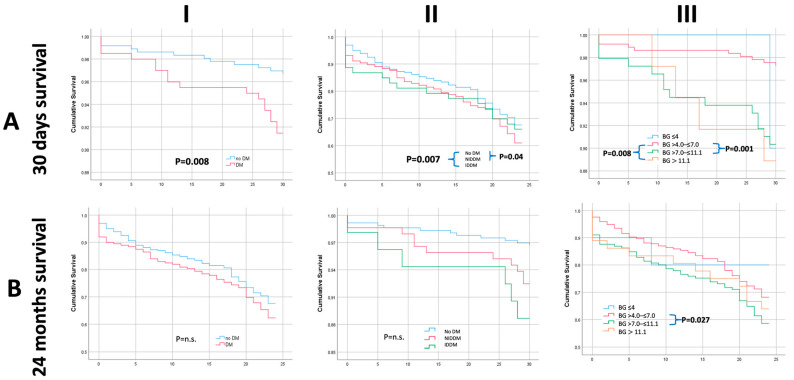
Kaplan–Meier univariate survival curves displaying short-term (30-day) mortality (**A**) and long-term (24-month) mortality (**B**). DM vs.no DM (**I**), no DM vs. NIDDM and IDDM (**II**), the four groups of admission fasting BG (**III**) BG: fasting blood glucose, DM: diabetes mellitus, IDDM: insulin-dependent diabetes mellitus, NIDDM: noninsulin-dependent diabetes mellitus.

**Table 1 life-13-00054-t001:** Characteristics of the patients by their DM status.

	All Patients N = 560	No DM N = 357	All DM N = 203	NIDDM N = 150	IDDM N = 53	*p*-Value *
Age (years) mean ± SD	79.97 ± 6.2	80.63 ± 6.19	79.97 ± 6.2	79.3 ± 6.19	77.45 ± 6.09	<0.001
Male gender (%)	41.07%	37.54%	47.29%	48.67%	43.4%	0.063
BMI (kg/m^2^) mean ± SD	27.85 ± 5.52	27.03 ± 5.13	29.31 ± 5.9	29.22 ± 6.12	29.6 ± 5.26	0.003
EuroSCORE II mean ± SD	5.73 ± 5.68	5.43 ± 5.31	6.27 ± 6.24	5.89 ± 5.85	7.35 ± 7.24	0.172
STS mortality score	5.01 ± 3.84	4.75 ± 3.87	5.46 ± 3.77	5.39 ± 3.85	5.65 ± 3.56	0.008
NYHA class mean ± SD	2.93 ± 0.76	2.93 ±0.77	2.93 ±0.73	2.91 ±0.74	2.98 ±0.74	0.83
Pre-TAVI-EF (%) mean ± SD	53.96 ± 16.12	54.27 ± 16.49	53.4 ± 15.47	52.66 ± 16.1	55.54 ± 13.41	0.481
Pre-TAVI aortic valve max gradient (mmHg) mean ± SD	79.25 ± 25.65	81.56 ± 26.1	75.23 ± 24.38	75.82 ± 25.1	73.57 ± 22.37	0.017
Pre-TAVI aortic valve mean gradient (mmHg) mean ± SD	49.74 ± 16.61	51.34 ± 17.13	46.96 ± 15.3	47.19 ± 15.51	46.32 ± 14.8	0.015
Pre-TAVI aortic valve area (cm^2^) mean ± SD	0.55 ± 0.16	0.55 ± 0.17	0.55 ± 0.15	0.54 ± 0.16	0.58 ± 0.14	0.22
Fasting blood glucose level on admission (mmol/L) mean ± SD	6.99 ± 2.53	6.13 ± 1.16	8.51 ± 3.42	8.34 ± 3.23	9.01 ± 3.88	<0.001
Heart rate on admission (/min) mean ± SD	74.81 ± 14.52	74.51 ± 14.17	75.32 ± 15.12	74.75 ± 12.67	76.94 ± 20.59	0.914
Creatinine on admission (ųmol/L) mean ± SD	113.89 ± 39.85	109.56 ± 36.43	121.49 ± 44.3	124.15 ± 45.89	113.95 ± 38.85	0.001
eGFR on admission (ml/min) mean ± SD	49.25 ± 11.27	50.34 ± 10.51	47.31 ± 12.29	46.86 ± 12.58	48.62 ± 11.44	0.036
White blood cell count (G/L) mean ± SD	8.1 ± 6.27	8.0 ± 6.72	8.28 ± 5.39	8.27 ± 6.07	8.31 ± 2.74	0.035
CRP on admission (mg/L) mean ± SD	8.43 ± 13.59	9.42 ± 15.18	6.71 ± 10.02	5.62 ± 7.53	10.0 ± 14.9	0.101
Triglyceride level on admission (mmol/L) mean ± SD	1.57 ± 0.92	1.5 ± 0.96	1.7 ± 0.84	1.72 ± 0.79	1.64 ± 0.99	0.001
Total cholesterol on admission (mmol/L) mean ± SD	4.54 ± 3.26	4.56 ± 1.33	4.51 ± 5.12	4.64 ± 5.85	4.12 ± 1.16	<0.001
Hypertension	90.34%	88.52%	93.56%	91.28%	100%	0.028
Atrial Fibrillation	42.22%	40.62%	45.05%	44.3%	47.17%	0,557
Coronary artery disease	70.48%	68.35%	74.26%	71.81%	81.13%	0.115
Prior PCI	32.02%	29.41%	36.63%	36.24%	37.74%	0.209
Prior CABG	11.45%	8.4%	16.83%	14.77%	22.64%	0.003
Prior myocardial infarction	16.46%	13.73%	21.29%	18.12%	30.19%	0.009
Porcelain Aorta	3.76%	3.64%	3.96%	4.03%	3.77%	0.943
Previous aortic valve implantation	2.15%	1.68%	2.97%	2.68%	3.77%	0.335
Peripheral arterial disease	10.73%	8.96%	13.86%	13.42%	15.09%	0.188
Carotid disease	30.05%	28.29%	33.17%	31.54%	37.74%	0.337
Previous stroke (CVA)	7.51%	6.72%	8.91%	7.38%	13.21%	0.247
COPD	17.95%	19.44%	15.35%	16.11%	13.21%%	0.431
History of PM/ICD implantation	11.96%	11.76%	12.32%	12.67%	11.32%	0.949

* *p*-values present the differences between the three groups (without DM, NIDDM, IDDM) BMI: body mass index, CABG: coronary artery bypass grafting, COPD: chronic obstructive pulmonary disease, CRP: high sensitivity C-reactive protein, DM: diabetes mellitus, EF: ejection fraction, eGFR: estimated glomerular filtration rate, IDDM: insulin-dependent diabetes mellitus, NIDDM: noninsulin-dependent diabetes mellitus, NYHA: New York Heart Association, PCI: percutaneous coronary intervention, TAVI: transcatheter aortic valve implantation.

**Table 2 life-13-00054-t002:** Clinical outcomes, procedural characteristics, and technical details by DM status.

	All patients N = 560	No DM N = 357	All DM N = 203	NIDDM N = 150	IDDM N = 53	*p*-Value *
30-day mortality	5.18%	3.08%	8.87%	8%	11.32%	0.008
1-year mortality	16.61%	14.85%	19.7%	19.33%	20.75%	0.323
2-year mortality	34.29%	32.21%	37.93%	39.33%	33.96%	0.304
Pre-TAVI dilatation	13.86%	15.59%	10.95%	12.84%	5.66%	0.133
Post-TAVI dilatation	22.97%	25.71%	18.23%	20.67%	11.32%	0.05
need for second valve implantation	2.88%	3.41%	1.97%	2%	1.89%	0.79
Post-procedural bleeding	19.21%	19.77%	18.23%	19.33	15.09%	0.722
Vascular complication	17.3%	16.38%	18.91%	19.59%	16.98%	0.685
Need for Vascular surgery	3.58%	3.94%	2.96%	2.67%	3.77%	0.774
Need for transfusion	29.26%	29.1%	29.56%	32.67%	20.75%	0.26
Tamponade during/after TAVI	2.34%	2.26%	2.48%	2.01%	3.77%	0.693
Need for emergency cardiac surgery	1.97%	1.94%	1.97%	1.33%	3.77%	0.5
Need for PM implantation	17.32%	18.49%	15.27%	18%	7.55%	0.141
New Renal insufficiency after TAVI	3.05%	2.54%	3.94%	4.67%	1.89%	0.443
Major Stroke after TAVI	1.97%	1.69%	2.49%	2.67%	1.89%	0.717

* *p*-values present the differences between the three groups (without DM, NIDDM, IDDM) DM: diabetes mellitus, IDDM: insulin-dependent diabetes mellitus, NIDDM: noninsulin-dependent diabetes mellitus, TAVI: transcatheter aortic valve implantation.

**Table 3 life-13-00054-t003:** Multivariate survival models. Short-term (30 days) mortality (A) and long-term (24 months) mortality (B).

A: 30-Day Mortality				B: 24-Month Mortality			
	Hazard Ratio	*p*-Value	Confidence Interval		Hazard Ratio	*p*-Value	CI
DM: yes	5.378	0.024	1.24–23.25	DM: yes	0.905	0.662	0.58–1.42
Age + 1	0.990	0.848	0.9–1.09	Age + 1	1.031	0.072	0.99–1.07
BMI + 1	0.966	0.568	0.86–1.09	BMI + 1	0.987	0.503	0.95–1.03
Pre-TAVI EF (%) + 1%	1.011	0.614	0.97–1.05	Pre-TAVI EF (%) + 1%	0.986	0.042	0.97–0.99
Pre-TAVI aortic mean gradient (mmHg) + 1	1.003	0.874	0.96–1.05	Pre-TAVI aortic valve mean gradient (mmHg) + 1	0.997	0.637	0.98–1.01
Pre-TAVI kreatinin (Umol/L) + 1	0.993	0.399	0.98–1.01	Pre-TAVI kreatinin (Umol/L) + 1	1.002	0.218	0.99–1.01
BG (mmol/L) + 1	1.088	0.332	0.92–1.29	BG (mmol/L) + 1	1.040	0.347	0.96–1.13
CRP (mg/L) + 1	1.014	0.473	0.98–1.05	CRP (mg/L) + 1	1.019	0.003	1.01–1.03
IDDM: yes	0.629	0.670	0.75–5.31	IDDM: yes	0.937	0.859	0.46–1.91
Age + 1	0.983	0.741	0.89–1.09	Age + 1	1.031	0.071	0.99–1.07
BMI + 1	0.990	0.866	0.88–1.11	BMI + 1	0.986	0.449	0.95–1.02
Pre-TAVI EF (%) + 1%	1.008	0.713	0.97–1.05	Pre-TAVI EF (%) + 1%	0.986	0.043	0.97–0.99
Pre-TAVI aortic mean gradient (mmHg) + 1	0.998	0.933	0.96–1.04	Pre-TAVI aortic valve mean gradient (mmHg) + 1	0.997	0.657	0.98–1.01
Pre-TAVI kreatinin (Umol/L) + 1	0.995	0.543	0.98–1.01	Pre-TAVI kreatinin (Umol/L) + 1	1.002	0.239	0.998–1.01
BG (mmol/L) + 1	1.208	0.015	1.04–1.41	BG (mmol/L) + 1	1.034	0.394	0.96–1.03
CRP (mg/L) + 1	1.007	0.730	0.97–1.04	CRP (mg/L) + 1	1.019	0.002	1.01–1.03

BMI: body mass index, BG: fasting blood glucose, CRP: high sensitivity C-reactive protein, DM: diabetes mellitus, EF: ejection fraction, eGFR: estimated glomerular filtration rate, IDDM: insulin-dependent diabetes mellitus, NIDDM: noninsulin-dependent diabetes mellitus, TAVI: transcatheter aortic valve implantation.

## Data Availability

The data that support the findings of this study are available from the first author (G.D) upon reasonable request due to privacy restrictions.

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
