# Peer review of "Elevated Fasting Glucose and C-Reactive Protein Levels Predict Increased All-Cause Mortality after Elective Transcatheter Aortic Valve Implantation"

_life, 2022, doi:10.3390/life13010054_

Round 1

Author Response

On behalf of the author team of our article entitled “Elevated Fasting Glucose and C-reactive Protein Levels Predict All-Cause Mortality After Elective Transcatheter Aortic Valve Implantation” I would like to thank the Reviewer for his thoughtful comments. Allow us to address your suggestions, and to incorporate them into our revised manuscript, as outlined below:

Suggestions:

”Clinical summary should be more concise and English editing should be considered”.

We have shortened our discussion section, and we have added a very brief and concise summary and conclusion section. Following your kind guidance, we had our manuscript reviewed by a native English speaker.

”In the title the connection between fasting glucose , CRP and short term mortality should be clarified”

Response: We greatly appreciate the Reviewer’s insightful comment. The title of the reviewed manuscript is changed accordingly: Elevated Fasting Glucose and C-reactive Protein Levels Predict Increased All-Cause Mortality After Elective Transcatheter Aortic Valve Implantation.

”Short conclusion should be drawn.”

As above mentioned it has been incorporated - thank you for the suggestion.

Notes:

“Line 112: Only T2DM patients were included. Later there is a subgroup of T1DM patients”

Answer: We agree that there is potential for confusion. In our study, all included patients suffered from type 2 diabetes (both insulin independent and insulin dependent). All diabetics were further divided into only orally treated, non-insulin dependent diabetes mellitus (NIDDM) and insulin dependent diabetes mellitus (IDDM). If their treatment included any insulin at all they were grouped into IDDM. In the revised manuscript table 2 the terms OTDM and ITDM are replaced by IDDM and NIDDM. Further, we had indeed used the term T1DM, but solely to state that our results may not apply to type I DM, see study limitations. The introduction section has been updated in the revised manuscript to clarify all as per your kind suggestion.

Questions:

”Pre TAVI coronary assessment was performed?”

Answer: We appreciate this very relevant question. In fact, all patients underwent a coronary assessment before TAVI. with the goal of revascularizing clinically relevant lesions (LM, proximal LAD or subtotal LAD, dominant CX, RCA, etc) before valve implantation. The majority of patients had invasive coronarography. However, in the last years of our data collection, certain low risk patients (for coronary artery disease) were studied with CT angiography to exclude clinically relevant lesions.

” If yes, then pre TAVI residual Syntax score was calculated for Nondiabetic, 2TDM, IDDM patients?”

We agree that this would have been an interesting suggestion but all significant coronary lesions in our study group were treated percutaneously prior to TAVI. Regrettably Syntax scores were not calculated.

”Diabetes is represented in all pre surgery scores. Should there be more weight on DM?”

Response: We completely agree with the Reviewer regarding the importance of DM as a risk factor for TAVI outcome. It is a goal of our study to emphasise the role of DM and to integrate this important risk factor into future risk scoring systems for TAVI candidates, similar to its previous integration into surgical risk scoring systems.

”Post TAVI dilatation rate is significantly higher in the non-diabetic patient group. Why? ”

This question is highly relevant. As highlighted in Table 1, patients with DM had significantly lower peak and mean aortic valve gradients than those without DM (75.2 mmHg vs. 81.6 mmHg and 47 mmHg vs. 51.3 mmHg, all p<0.05). More severe AS obviously more comm only needs pre- and post-dilatation, although in our cohort this proved to be statistically significant only for post dilatation (see Table 2). We hypothesize that patients with DM usually have closer medical follow up, so that their AS is detected earlier, as also evidenced by their significantly younger age in our study (80.6 vs.79.9 p<0.001).

”Vascular complication rate is high overall, so the question comes up, that how would you describe vascular complication”.

Response: Again thank you very much for this question. Our vascular complication percentages include minor as well as major vascular complications based on the VARC2 criteria. Furthermore it is our Institutional policy to avoid alternative accesses since their mortality rate proved to be higher than the transfemoral approach, thereby accepting many patients with sub optimal femoral systems and incurring slightly more peripheral (but non-lethal) complications.

We greatly appreciate the Reviewer’s suggestions and are convinced that they have helped us make our paper better. Should there be additional comments or suggestions, we would welcome them.

Reviewer 2 Report

The authors submitted a research article in which they elucidated the impact of a series of risk-factor candidates on post-TAVI mortality . They included 560 consecutive elective elderly patients with severe symptomatic AS and either prohibitive or extremely high surgical risk in retrospective single-center observational study. The authors followed the patients to evaluate all-cause mortality at 30 days and 2 years post TAVI-intervention. They established that not only a history of DM, but also an elevated admission fasting BG unrelated to the presence or absence of a patient-history of DM, and, independently, CRP levels have prognostic value. The aim of the study is clear. The paper contains weel-written subsections and has clear logical structure. The tables and figures a re clear and legible. The conclusive part is comprehensive and clear. Although the authors received intriguing evidence regarding hypothesis, I would like to put forward several commenst to discuss.

1. The authors indicated that all patients had no medical treatment, but only TAVI procedures, while a signature of comorbidities among elder people requires concomitant medication. Whether a lack of personally adjusted therapy is a cause of clinical outcomes in the study, but not direct impact of TAVI ?

2. The avarage of CRP levels among the patients was less 10 mg/L, whereas upper quartiles vs lower quartiles might be qualified as plausible co-factor to intervene in the outcomes. Please, check and comment.

3. Please, add clear explanation of abbreviations included in Figures and Tables.

4. The patients characteristics are reported with a focus on diabetes mellitus, but not other concomitant disease. Please, add this information and interprete it.

Author Response

On behalf of the authors of our article entitled “Elevated Fasting Glucose and C-reactive Protein Levels Predict All-Cause Mortality After Elective Transcatheter Aortic Valve Implantation” I would like to thank the Reviewer for his thoughtful comments. Allow us to address your suggestions, and to incorporate them into our revised manuscript, as outlined below:

  1. The authors indicated that all patients had no medical treatment, but only TAVI procedures, while a signature of comorbidities among elder people requires concomitant medication. Whether a lack of personally adjusted therapy is a cause of clinical outcomes in the study, but not direct impact of TAVI”

Response: We agree that our paper, as worded previously, may have caused some misunderstanding. What we intended to say is that, at present, no universally accepted medical treatment for AS exists. As properly noted by the reviewer, all of our patients were on medical therapy for other concomitant diseases before and after the procedure. Our revised text has been changed accordingly in the method section .

  1. The average of CRP levels among the patients was less 10 mg/L, whereas upper quartiles vs lower quartiles might be qualified as plausible co-factor to intervene in the outcomes. Please, check and comment. ”

Response: In the descriptive statistics section of the manuscript, CRP values are presented using mean and standard deviation (all patients: 8.43±13.59, no DM: 9.42±15.18, all DM: 6.71±10.02; OTDM: 5.62±7.53, ITDM: 10.0±14.9).
These data do not necessarily overlap with the medians and hence the interquartile range. The measured CRP data are spread over a wide range (median: 3.7; interquartile range:1.4 - 8.8) therefor, for ease of interpretation and because all other continuous data are presented with means and standard deviations, this same method is used here.
Further, in the multivariate model, we adjusted for the effect of CRP as a continuous variable, showing a significant association with 24-month mortality but no association with 30-day mortality. 

  1. Please, add clear explanation of abbreviations included in Figures and Tables.

Thank you very much this comment, a list of abbreviations has been added to the revised manuscript; further, in accordance with the Reviewer’s suggestion, all three Tables and the Figure are now supplied with the respective abbreviations.

  1. The patients characteristics are reported with a focus on diabetes mellitus, but not other concomitant disease. Please, add this information and interprete it.

We agree with the Reviewer that the presence of concomitant conditions and diseases constitutes important information. We tried to display all relevant concomitant diseases in table 1. The main focus of our article was to assess the impact of DM  (both insulin- and non insulin dependent) and of elevated fasting blood glucose levels on the outcome of transcatheter valve implantation. Multivariate adjustment were made in order to reveal their independent significance predicting patients’ outcome. In our model no variable selection was performed, every variable was entered into the multivariate model without univariate prefiltering, as this is known to introduce bias. Variables were selected based on their expected importance in post-TAVI mortality and DM. In our previous multivariate Cox regression analysis (Table 3) diabetes status and insulin treated diabetes were tested in separate models, but seven other factors (Age, BMI, pre TAVI EF, pre TAVI aortic mean gradient, pre TAVI creatinine, fasting blood glucose, CRP) were the same in the two models. These factors were selected based on their assumed importance. In univariate analysis (see Table 1) six out of seven (age, BMI, pre TAVI aortic mean gradient, pre TAVI creatinine, admission plasma glucose and CRP) proved to be statistically different between the groups. In accordance with the Reviewer’s comment a univariate Cox-analysis has now been performed:

 30-day mortality

2-year mortality

Hazard Ratio

P-value

95% confidence interval

Hazard Ratio

P-value

95% confidence interval

EuroSCORE II

1,0157

0,686

0,942

1,096

1,0193

0,157

0,993

1,047

STS mortality score

1,0593

0,071

0,995

1,128

1,0350

0,052

1,000

1,072

Age

0,9484

0,056

0,898

1,001

1,0243

0,075

0,998

1,052

BMI

0,9950

0,925

0,898

1,103

0,9734

0,122

0,941

1,007

Female gender

0,5406

0,144

0,237

1,233

0,7402

0,054

0,545

1,005

fasting blood glucose

1,1373

0,017

1,024

1,264

1,0267

0,368

0,970

1,087

DM

2,8969

0,013

1,254

6,693

1,0582

0,728

0,770

1,455

NIDDM

2,5201

0,050

1,000

6,349

1,1475

0,432

0,814

1,618

IDDM

3,9638

0,014

1,328

11,828

0,8110

0,492

0,446

1,473

NYHA class

1,0484

0,864

0,611

1,800

1,0207

0,845

0,832

1,253

Pre TAVI EF

0,9947

0,679

0,970

1,020

0,9870

0,007

0,978

0,996

Pre TAVI aortic valve max gradient

0,9952

0,571

0,979

1,012

0,9922

0,017

0,986

0,999

Pre TAVI aortic valve mean gradient

0,9968

0,802

0,972

1,022

0,9875

0,014

0,978

0,997

Pre TAVI aortic valve area

1,2598

0,874

0,072

21,821

1,3844

0,513

0,523

3,665

Pre TAVI creatinine

1,0036

0,396

0,995

1,012

1,0039

0,009

1,001

1,007

Triglyceride level on admission

0,7229

0,354

0,364

1,435

1,0637

0,441

0,909

1,245

Total cholesterol on admission

0,7362

0,095

0,514

1,055

1,0134

0,391

0,983

1,045

GFR on admission

0,9842

0,363

0,951

1,019

0,9823

0,006

0,970

0,995

White blood cell count

0,9569

0,568

0,823

1,113

1,0145

0,101

0,997

1,032

CRP on admission

1,0271

0,001

1,011

1,043

1,0076

0,156

0,997

1,018

Hypertension

1,1248

0,874

0,264

4,797

1,3305

0,323

0,755

2,344

Atrial Fibrillation

1,0417

0,923

0,457

2,376

0,9726

0,861

0,713

1,326

Peripheral arterial disease

1,1769

0,732

0,464

2,985

1,1388

0,461

0,806

1,609

Prior PCI

1,3921

0,439

0,603

3,216

1,3014

0,103

0,948

1,786

Prior myocardial infarction

1,0738

0,897

0,365

3,156

1,1186

0,584

0,749

1,671

Prior CABG

0,7327

0,674

0,172

3,125

0,9801

0,936

0,601

1,599

Porcelain Aorta

2,6300

0,191

0,617

11,217

0,4680

0,193

0,149

1,467

Previous aortic valve implantation

2,2986

0,416

0,310

17,054

0,5989

0,471

0,149

2,415

Carotid disease

0,4791

0,181

0,163

1,408

1,4435

0,022

1,054

1,977

Previous stroke

NA

NA

NA

NA

1,0978

0,738

0,635

1,899

COPD

0,9676

0,952

0,329

2,844

1,3072

0,165

0,896

1,907

History of PM/ICD implantation

1,5496

0,426

0,527

4,555

0,9328

0,775

0,578

1,505

Need for PM implantation post-TAVI

1,6828

0,273

0,663

4,268

1,5525

0,018

1,077

2,239

Need for emergency cardiac surgery

7,9294

0,005

1,858

33,833

0,5345

0,532

0,075

3,818

Need for Vascular surgery

2,6135

0,194

0,613

11,147

0,6688

0,427

0,248

1,804

need for second valve implantation

3,4585

0,094

0,811

14,751

0,9769

0,963

0,362

2,635

Pre-TAVI dilatation

0,9372

0,917

0,278

3,154

1,8759

0,001

1,290

2,729

Post-TAVI dilatation

2,1991

0,065

0,952

5,081

0,9802

0,916

0,677

1,419

Vascular complication

1,7187

0,254

0,678

4,359

1,3247

0,144

0,908

1,932

Need for transfusion

2,2832

0,048

1,007

5,174

1,3896

0,045

1,007

1,918

Tamponade during/after TAVI

6,0008

0,015

1,407

25,599

0,7911

0,742

0,196

3,190

New Renal insufficiency after TAVI

22,5678

0,000

9,534

53,422

1,7654

0,262

0,655

4,761

Major Stroke after TAVI

5,0300

0,029

1,179

21,453

2,8170

0,023

1,156

6,864

Out of the 8 factors that had previously been entered into the multivariate model, fasting blood glucose, NIDDM, IDDM, CRP level, pre-TAVI, creatinine, and pre-TAVI EF, were proven to be significant in our univariate analysis.

We greatly appreciate the Reviewer’s suggestions and are convinced that they have helped us make our paper better. Should there be additional comments or suggestions, we would welcome them.

Round 2

Reviewer 2 Report

The authors submitted a revised version of the paper along with a clear description of the ways by which they improved the article. I have no serious concerns about the article in its revised version.